# The Effect of Aldosterone on Cardiorenal and Metabolic Systems

**DOI:** 10.3390/ijms24065370

**Published:** 2023-03-11

**Authors:** Hiromasa Otsuka, Masanori Abe, Hiroki Kobayashi

**Affiliations:** 1Department of Internal Medicine, Hatogaya Hospital, Saitama 334-0003, Japan; 2Department of Emergency Room and General Medicine, Ageo Central General Hospital, Saitama 362-8588, Japan; 3Division of Nephrology, Hypertension, and Endocrinology, Department of Medicine, Nihon University School of Medicine, Tokyo 173-8610, Japan

**Keywords:** primary hyperaldosteronism, vascular system, metabolic alterations, oxidative stress, thrombosis

## Abstract

Aldosterone, a vital hormone of the human body, has various pathophysiological roles. The excess of aldosterone, also known as primary aldosteronism, is the most common secondary cause of hypertension. Primary aldosteronism is associated with an increased risk of cardiovascular disease and kidney dysfunction compared to essential hypertension. Excess aldosterone can lead to harmful metabolic and other pathophysiological alterations, as well as cause inflammatory, oxidative, and fibrotic effects in the heart, kidney, and blood vessels. These alterations can result in coronary artery disease, including ischemia and myocardial infarction, left ventricular hypertrophy, heart failure, arterial fibrillation, intracarotid intima thickening, cerebrovascular disease, and chronic kidney disease. Thus, aldosterone affects several tissues, especially in the cardiovascular system, and the metabolic and pathophysiological alterations are related to severe diseases. Therefore, understanding the effects of aldosterone on the body is important for health maintenance in hypertensive patients. In this review, we focus on currently available evidence regarding the role of aldosterone in alterations of the cardiovascular and renal systems. We also describe the risk of cardiovascular events and renal dysfunction in hyperaldosteronism.

## 1. Introduction

Aldosterone, a hormone that regulates sodium balance and blood pressure in humans, has also various pathophysiological roles. Overproduction of aldosterone, also known as primary aldosteronism (PA), is the most common secondary cause of hypertension, and the prevalence of PA in patients with hypertension is 5–15% [1,2]. It is also associated with cardiovascular and renal complications, as well as metabolic alterations, such as insulin intolerance and obesity [3,4,5]. Moreover, mortality due to cardiovascular complications is higher in patients with PA than in those with essential hypertension (EH) [6]. Thus, the excess of aldosterone can have adverse systemic effects and lead to poor prognosis.

The excess of aldosterone induces pathophysiological alterations such as exacerbated inflammation, oxidative stress, and fibrosis. These altered mechanisms are related to each other and cause metabolic abnormality, coronary artery disease (CAD), and renal dysfunction [1,6,7]. Especially, vascular and structural changes in the heart are reported to increase the risk of cardiovascular disease and mortality. In fact, excess aldosterone induces vascular stiffness, thrombosis, and left ventricular remodeling, and enhances mortality rate [8,9]. Aldosterone can exert its functions through its interaction with the mineralocorticoid receptor (MR). Therefore, MR antagonists can reduce cardiovascular disease and mortality as well as suppress the progression of renal dysfunction [10]. MR antagonists such as spironolactone and eplerenone are well-evidenced medications for resistant hypertension with PA and heart failure with reduced ejection fraction [10,11], and can prevent the effect of excess aldosterone and reduce the risk of cardiovascular morbidity and mortality.

In this review, we focus on the current evidence of pathophysiological alterations of the metabolic and cardiovascular systems by the excess of aldosterone. The physiology of aldosterone is also described. In this context, we discuss the connection between these aldosterone-induced alterations, especially in PA, and the risk of cardiovascular disease (CVD). Moreover, due to the association between renal damage and alterations in the cardiovascular system, we also aim to describe the association between aldosterone and the renal system.

## 2. Aldosterone Synthesis and Physiology

### 2.1. Basic Mechanism of Aldosterone Synthesis and Physiological Processes

Aldosterone is a mineralocorticoid, a class of steroid hormones. It raises blood pressure and regulates the water–mineral balance in the body [12,13]. It is mainly produced in the zona glomerulosa of the adrenal cortex by the aldosterone synthase (CYP11B2), which belongs to the cytochrome P450 family [14,15]. Aldosterone synthesis is mainly stimulated by extracellular potassium concentrations, adrenocorticotropic hormone, and the renin-angiotensin system (RAS) [16]. It is synthesized from cholesterol, which is transported to the inner mitochondrial membrane. Cytochrome P450scc (cholesterol side-chain cleavage, encoded by CYP11A1) converts cholesterol to pregnenolone [17,18], which is transported to the smooth endoplasmic reticulum and converted to 11-deoxycorticosterone (11DCS) by 3β-hydroxysteroid dehydrogenase and 21-hydroxylase. In the mitochondria, 11DCS is 11-hydroxylated, 18-hydroxylated, and 18-oxidated continuously by aldosterone synthase to produce aldosterone [19]. Aldosterone easily diffuses through the cell membrane and binds to MR in the cytoplasm. After the binding of aldosterone to MR, the receptor is dimerized, translocates to the nucleus, and acts as a ligand-activated transcription factor that leads to the expression of genes that encode proteins such as serum and glucocorticoid-stimulated kinase 1 (SGK-1), which regulates and activates ionic transport proteins such as the epithelial cell sodium channel (ENaC) in principal cells [20]. Activated ENaC promotes sodium and water reabsorption and regulates fluid volume and sodium balance [21]. Sodium homeostasis is regulated by proton pumps, such as H^+^-ATPase and H^+^-K^+^-ATPase, in intercalated cells [22,23].

Glucocorticoid (GC) is also a steroid hormone that binds to MR, which is determined by the balancing of expression levels of 11β hydroxysteroid dehydrogenase types 1 and 2 (11βHSD1 and 2) [24]. These enzymes act as a prereceptor gateway expressed in most cells, although the expression levels differ, e.g., 11βHSD1 is highly expressed in the liver, central nerve, and smooth muscle, while 11βHSD2 is highly expressed in the renal distal nephron and colon. Cardiomyocyte and inflammatory cells are not expressed 11βHSD2 [24,25]. 11βHSD1 reactivates intracellular GC via reductase activity, while 11βHSD2 decomposes GC via dehydrogenase activity, suggesting a selectivity for binding of aldosterone to MR in cells.

The gene translational pathway activation through MR is associated with the genomic effect of aldosterone, whereas the non-gene translational pathway is associated with the non-genomic effects of aldosterone characterized by the rapid vascular response through direct aldosterone actions or MR [26]. G protein-coupled estrogen receptor 1 (GPER1) has been proposed as a likely aldosterone receptor, and it may contribute to the rapid effects of aldosterone in several tissues. This receptor is located in the cell membrane. GPER1 is expressed in endothelial cells, vascular smooth muscle cells, and cardiomyocytes [27,28]. GPER1 is also expressed in tubular epithelial cells, mesangial cells, and renal interlobular arteries in the kidney [29,30,31]. GPER1 activation is associated with protective effects in the vasculature and with the regulation of cell growth, migration, and cell death. Moreover, GPER1 contributes to the rapid activation of the extracellular signal-regulated kinase (ERK) and apoptosis, which is induced by aldosterone in vascular smooth muscle cells. However, it remains unclear whether GPER1 is a receptor for aldosterone or not. The non-genomic effects of aldosterone are also modulated by other membrane receptors such as the epidermal growth factor receptor (EGFR), platelet-derived growth factor receptor (PDGFR), insulin-like growth factor 1 receptor (IGF1R), and angiotensin II receptor type 1 (AT1). EGFR, PDGFR, and IGF1R are transactivated via receptor tyrosine kinases, while AT1 is a G protein-coupled receptor, like GPER1. Scaffolding proteins, such as striatin (STRN) and caveolin-1 (CAV1), facilitate MR attachment to the cell membrane, and the aldosterone–MR complex interacts with these receptors and elicits non-genomic effects [32].

### 2.2. Pathophysiological Effect of Aldosterone

As mentioned above, aldosterone is an essential hormone that maintains electrolytes and fluid homeostasis. However, the homeostasis collapses when aldosterone is pathologically in excess, such as in PA. PA is a clinical syndrome caused by autonomous aldosterone overproduction and suppressed plasma renin, which leads to hypertension, hypokalemia, and metabolic alkalosis. Major forms of PA are unilateral (usually aldosterone-producing adenoma) and bilateral (bilateral adrenal hyperplasia) [33]. The screening test of PA is performed to determine the ratio of plasma aldosterone concentration (ng/dL) to plasma renin activity (ng/mL/h), which is expressed as the aldosterone-to-renin ratio (ARR). The test is positive when ARR is greater than 20 (ng/dL)/(ng/mL/h). The patients with a positive screening result need confirmatory tests such as the captopril challenge test and saline infusion test to prove autonomous aldosterone production. For example, a positive result is that ARR is greater than 20 (ng/dL)/(ng/mL/h) at 60 or 90 min after 50 mg of captopril intake orally in the captopril challenge test [34]. Thus, clinical abnormality, such as hypertension and hypokalemia caused by aldosterone, needs higher aldosterone concentration in the plasma. However, it may not become an abnormal clinical status to be plasma aldosterone only, as was stated in an interesting report about New Guinea inhabitants. People in the highlands of New Guinea have a very low sodium intake because the staple diet consists of yams, which have a low sodium content. They have low normal blood pressure and prominently high levels of plasma aldosterone. The plasma aldosterone levels in this population are much higher than those in patients with PA. Interestingly, these people in New Guinea have no cardiovascular damage [35]. In animal models, aldosterone causes cardiovascular and renal injury only with an inappropriate intake of salt [36]. Thus, aldosterone is a risk factor for CVD when the physiological aldosterone feedback is disturbed and the individual has a high salt intake.

## 3. Metabolic and Other Pathophysiological Alterations Caused by Aldosterone 

### 3.1. Aldosterone and Metabolic Alterations

Aldosterone levels are associated with insulin resistance and visceral obesity. Some reports described sex differences, whereas other studies found none [37,38,39]. Several mechanisms underlie insulin resistance and visceral obesity. Aldosterone induces impaired vasodilation and limits insulin signaling [40]. Aldosterone treatment of cultured vascular smooth muscle cells increases the proteasomal degradation of the insulin receptor substrate-1 (IRS-1) and attenuates insulin-induced Akt phosphorylation and glucose uptake. This prevents the activation of Src kinase, which decreases the expression of IRS-1 and the generation of reactive oxygen species (ROS) induced by the MR antagonist. Therefore, aldosterone-induced insulin resistance is caused by excessive serine phosphorylation of IRS-1 [41]. Moreover, aldosterone induces insulin resistance in skeletal muscles. In a rat model, aldosterone treatment impaired the rate of glucose uptake, glucose oxidation, and insulin signal transduction in the gastrocnemius muscle via reduced expression of IRS-1, Akt, and plasma membrane glucose transporter (GLUT) 4 genes [42]. Aldosterone also impairs glucose uptake in adipose cells. Several studies have demonstrated that impaired glucose uptake in adipose cells by aldosterone reduces cell surface localization of GLUT4, as well as IRS-1, phosphoinositide 3-kinase, and Akt phosphorylation [43,44]. In another study, obesity-related insulin resistance was improved by an MR antagonist by reducing excessive ROS production [45]. A comparison between patients with PA and those with EH showed low C-peptide levels, together with a reduction of pancreatic beta-cell function indices in patients with PA after a homeostasis model assessment [46]. Thus, these findings suggest that aldosterone could influence beta-cell function, leading to impaired glucose metabolism.

Obesity is a common condition in individuals with high aldosterone levels [47]. In a preclinical study, obese mice with a high-fat diet had higher levels of aldosterone than lean mice [48]. However, this does not confirm that aldosterone excess promotes obesity. A pathophysiological explanation may be the leptin-related aldosterone overproduction in obese individuals. Leptin was proposed as one of the direct aldosterone-stimulating factors. Allegedly, it stimulates aldosterone synthesis because it can act directly on adrenal glomerulosa cells to increase aldosterone synthase expression and enhance aldosterone production via calcium-dependent mechanisms [49]. Interestingly, only female mice showed a marked increase in adrenal aldosterone synthase expression and plasma aldosterone levels by leptin [50]. Moreover, female mice treated with spironolactone showed reduced blood pressure compared to male mice. Leptin deficiency or impaired leptin signaling in mice or humans does not cause elevated plasma aldosterone levels. Endogenous high leptin levels or exogenous leptin supplementation increased aldosterone synthase expression and aldosterone production [49]. Therefore, obesity-induced high leptin levels play a central role in the elevation of aldosterone levels. Another factor related to aldosterone release is adipokine-like complement-C1q tumor necrosis factor-related protein 1 (CTRP1), which is released from adipose tissue [51]. The role of CTRP1 is to regulate the body’s energy homeostasis and sensitivity to insulin. CTRP1 affects the expression of CYP11B2 and stimulates aldosterone synthesis in the adrenal cortex [52]. CTPR1 is higher in individuals with obesity and increased in those with hypertension [51]. Increased CTRP1 is also associated with obesity and cardiovascular events [53]. However, plasma CTRP1 levels are not associated with plasma aldosterone levels in obese and non-obese patients with chronic kidney disease (CKD) [54]. Thus, the increase in CTRP1 in obesity remains controversial. Moreover, the role of CTRP1 in releasing aldosterone remains unclear. Further investigation is needed to clarify the association between CTRP1 and obesity for releasing aldosterone.

### 3.2. Oxidative Stress and Inflammation 

Aldosterone directly affects the vascular system through inflammation, oxidative stress, endothelial dysfunction, fibrosis, and hypertrophic remodeling [55,56]. These contribute to the occurrence of CAD, left ventricular (LV) hypertrophy, heart failure (HF), atrial fibrillation (AF), increased carotid intima–media thickness, and CVD [1,57,58,59]. Patients with PA have an increased risk of cardiovascular death compared to those with primary hypertension [6]. Thus, excess aldosterone induces harmful alterations in the cardiovascular pathophysiology and results in worse clinical outcomes. 

Aldosterone (Aldo) binds to the mineralocorticoid receptor (MR) and this complex induces oxidative stress by the production of reactive oxygen species (ROS) through the activation of nicotinamide adenine dinucleotide phosphate (NADPH) oxidase and mitochondria. Increased production of ROS promotes the activation of nuclear factor kappa-light-chain-enhancer of activated B cells (NF-kB), activator protein-1 (AP-1), and oxidized Ca^2+^/calmodulin-dependent protein kinase II (CaMK II) via NADPH oxidase activation. These factors contribute to inducing inflammation. This complex also stimulates the expression of profibrotic molecules, including plasminogen activator inhibitor-1 (PAI-1), transforming growth factor-β1 (TGF-β), endothelin 1 (ET-1), and connective tissue growth factor (CTGF). These molecules are known to increase the risk of fibrosis onset. Cortisone is converted to cortisol by the action of 11 β–hydroxysteroid dehydrogenase 1 (11β–HSD1) and binds to MR. The cortisol–MR complex contributes to oxidative stress. 

The Aldo–MR complex is attached to the cell membrane by caveolin–1 (CAV-1) and striatin, thereby phosphorylating of tyrosine receptors (TK) and extracellular signal-regulated protein kinases 1 and 2 (ERK1/2) occurs. TK, like the epidermal growth factor receptor (EGFR), and G-protein coupled receptors, like the G-protein estrogen receptor (GPER), crosstalk and interact with the Aldo–MR complex.

#### 3.2.1. Oxidative Stress 

The aldosterone–MR complex induces an increase in inflammation through oxidative stress (Figure 1 and Figure 2). Oxidative stress is an imbalance between the production of free radicals and antioxidant protection in the body. Free radicals are produced by normal biological processes such as breathing and digesting food. Especially, free radicals that contain oxygen are called reactive oxygen species (ROS). The ROS are generated by nicotinamide adenine dinucleotide phosphate (NADPH) oxidase. The ROS induces protein oxidation and dysregulates cell signaling, which leads to cell damage and death. It also contributes to inflammation and fibrosis. Aldosterone is also associated with ROS production and induces oxidative stress on the vascular system [55]. Aldosterone increases NADPH oxidase activity and oxidative stress in macrophages, endothelial cells, and cardiomyocytes [60,61,62]. In addition, aldosterone promotes cardiovascular inflammation by inducing the oxidation of Ca^2+^/calmodulin-dependent protein kinase II (CaMK II) via NADPH oxidase activation, and it contributes to LV remodeling after MI [63,64]. Aldosterone decreases the endothelial expression of glucose-6-phosphate dehydrogenase (G6PD). G6DP reduces oxidized NADPH to NADPH, which is utilized as a reducing equivalent to limit ROS levels [65]. Thus, aldosterone stimulation results in the accumulation of ROS due to the reduction of G6DP levels leading to decreased NADPH concentrations. These responses are attenuated by antioxidants and MR antagonists [60].

#### 3.2.2. Inflammation

Inflammation is a biological defense mechanism by the immune system which defends the body from harmful agents or stimuli such as pathogens and damaged tissues. First, recruiting immune cells produces inflammatory mediators which contribute to local inflammation. Usually, the inflammation is controlled and repairs the damage. Finally, remodeling and fibrosis occurs to heal the inflammation site. However, when the inflammation is exacerbated and chronic, it can cause serious tissue changes and fibrosis such as atherosclerosis and ventricular hypertrophy.

Aldosterone increases the expression of inflammatory adhesion molecules such as intracellular adhesion molecule-1 (ICAM-1) and vascular cell adhesion molecule-1. It leads to tissue infiltration of CD68-positive cells [60,66]. Other inflammatory markers such as cyclooxygenase-2 and monocyte chemoattractant protein 1 are highly expressed in the heart, vasculature, and kidney [66,67]. Aldosterone also stimulates macrophage infiltration and increases the production of ROS, such as superoxide and hydrogen peroxide. This triggers the activation of proinflammatory transcription factor activator protein (AP)-1 and nuclear factor kappa-light-chain-enhancer of activated B cells (NF-κB), which in turn induces the production of adhesion molecules and chemokines [68]. In addition, the infiltrating macrophages also act on oxidized low-density lipoproteins and form atherosclerotic plaque. As aldosterone induces endothelial dysfunction such as fibrosis, aldosterone-induced inflammation with atherosclerotic plaque in the endothelium causes vascular ischemia [8]. Fibrosis occurs when collagen and matrix production exceed their degradation by matrix metalloproteinases. Aldosterone stimulates the expression of profibrotic molecules, including plasminogen activator inhibitor-1 (PAI-1), TGF-β, endothelin 1, CTGF, placental growth factor, osteopontin, and galectin-3. Aldosterone increases the expression of PAI-1, which is a member of the serpin (serine protease inhibitor) superfamily and inhibits the activation of plasminogen to plasmin by the tissue plasminogen activator in clustered endothelial cells, cardiomyocytes, and monocytes [69,70]. In turn, PAI-1 induces fibrosis and remodeling by disturbing plasmin-mediated metalloproteinase activation and extracellular matrix degradation. Aldosterone promotes collagen secretion and synthesis from cardiac myocytes and fibroblasts through the activation of MRs, oxidative stress, and chronic inflammation [71]. Aldosterone activates mitogen-activated protein kinase (MAPK) pathways, promoting the proliferation of myofibroblasts that contribute to collagen deposition in the myocardium [72]. Aldosterone also increases the expression of TGF-β and CTGF via MR activation, and it stimulates the production of matrix proteins in the myocardium [64]. In the myocardium, CTGF expression increases via the activation of the G-protein-coupled receptor kinase 2 (GRK2) and results in collagen deposition.

## 4. Vascular Stiffness and Thrombosis

As mentioned above, increased aldosterone levels change metabolic and other pathophysiological effects in the body. Especially in the vasculature, the lumen is narrowed by inflammation and oxidative stress in the endothelium, and stenosis and obstruction can occur. Thus, hemodynamical changes or thromboembolic events can result in cardiovascular events. 

### 4.1. Aldosterone and Vascular Stiffness

Aldosterone activates ENaC thereby controlling sodium balance and blood pressure. Excess aldosterone enhances ENaC signaling, which plays an important role in the development of endothelial and vascular stiffness. ENaCs consist of three subunits: α, β, and g. The α/β subunit complex is essential for channel function. Endothelial ENaC α subunit knockout in mice prevents ENaC activation, endothelium stiffness, reduction of endothelial nitric oxide synthase (eNOS) activity, inflammation, oxidative stress, and aortic remodeling. Thus, physiological changes caused by enhanced ENaC signaling lead to endothelial stiffness, reduction of eNOS activity, inflammation, oxidative stress, and aortic remodeling [73].

### 4.2. Aldosterone and Thrombosis 

Aldosterone levels are positively correlated with an increased risk of acute cardiovascular thrombotic events [9]. This aldosterone effect is related to enhanced coagulation, reduced fibrinolysis, increased oxidative stress, and reduced nitric oxide bioavailability [74]. MR antagonists and angiotensin II receptor antagonists do not completely suppress the prothrombotic effect of aldosterone [75]. Therefore, the thrombosis mechanism through aldosterone suggests not only a genomic effect but also a non-genomic effect. The STRN and CAV1 interacts with MR to regulate the rapid non-genomic actions of aldosterone [76]. In a human study, individuals with a single nucleotide polymorphic gene variant of STRN, rs2540923, exhibited salt-sensitive blood pressure [77]. STRN heterozygote knockout (Strn^+/−^) mice also exhibited salt-sensitive blood pressure, mildly chronically elevated aldosterone levels, enhanced aortic vasoconstriction, decreased vascular relaxation, and reduced aortic eNOS expression [78]. 

Aldosterone administration promoted stasis-induced venous thrombosis in rats and laser- or FeCl_3_-induced thrombosis in mesenteric venules in mice [74,79]. The aldosterone mechanism of the prothrombotic action is related to its rapid and simultaneous effects on platelets, plasma- and endothelium-dependent hemostatic factors, and alterations in the clot structure, thereby making it resistant to fibrinolysis.

Coronavirus disease 2019 (COVID-19), which is caused by severe acute respiratory syndrome coronavirus 2 (SARS-CoV-2), is characterized by severe inflammation, RAS imbalance, and vascular coagulopathy [80]. In severe COVID-19 cases, hyperaldosteronism has been suggested based on the presence of hypokalemia [81]. Aldosterone levels correlate with PAI-1 levels, and aldosterone directly increases PAI-1 production [82,83]. Thus, hyperaldosteronism may impede fibrinolysis by overexpression of PAI-1, cause clot formation, and ultimately result in vascular thrombosis.

## 5. Aldosterone and Major Cardiovascular Disease 

### 5.1. Coronary Artery Disease

The prevalence of ischemic heart disease (IHD) is 2.1% in Japanese patients with PA [84]. Other ethnicities have similar CVD prevalence rates [6,85]. Patients with PA experienced more non-fatal MIs or angina than patients with EH [86]. The reason patients with PA have a high risk of IHD is not only that aldosterone affects the vasculature and cardiac muscle, but it also increases the risk of CVD through pathophysiological alterations such as metabolic and inflammatory changes that contribute to the development and progression of IHD. Table 1 shows the cardiovascular and renal disease associated with aldosteronism. 

### 5.2. Left Ventricular Remodeling and Heart Failure

LV remodeling is defined by LV hypertrophy and fibrosis. LV remodeling leads to LV dysfunction and results in HF [98]. In clinical studies, patients with PA have LV remodeling compared with patients with EH [99,100]. In animal studies, it was reported that aldosterone also stimulated LV rat myocytes directly, causing LV hypertrophy. Cardiotrophin-1 (CT-1) is a cytokine that induces cardiomyocyte hypertrophy, increases the expression of myosin light chain and skeletal α-actin, and enhances myosin light chain phosphorylation in rodents [101]. Moreover, increased myocardial expression of CT-1 has been found in aldosterone-infused mice with high salt intake. Furthermore, CT-1-null mice have been resistant to aldosterone-induced LV hypertrophy and fibrosis [102]. These findings support aldosterone-induced fibrosis leading to remodeling such as LV hypertrophy, impaired LV filling, and left atrial dilation. In addition, LV remodeling causes LV stiffness, which increases LV end-diastolic pressure (LVEDP), and elevated LVEDP can lead to decreased myocardial oxygen supply [103]. Thus, LV remodeling causes myocardial oxygen supply–demand mismatch, which leads to myocardial ischemia. 

### 5.3. Atrial Fibrillation

AF is a common disease in older adults, and its prevalence increases with age. In patients with PA, the risk of AF is reported to be 3.52 times higher than in patients with EH [1]. Another study demonstrated that patients with hypertension and unexplained AF have a high PA prevalence [92]. A prospective study found that patients with PA who underwent adrenalectomy had similar long-term outcomes as optimally treated patients with EH. However, medically treated patients with PA had worse AF-free survival rates than patients with EH. Aldosterone also promotes AF in rodents. Aldosterone leads to structural and functional changes, mainly characterized by atrial fibrosis, myocyte hypertrophy, and conduction disturbance [104]. Interestingly, patients with AF have higher expression of atrial MR compared with those with sinus rhythm [105]. In an experimental study in rodents, MR expression was increased with rapid depolarization through a calcium-dependent mechanism [98]. MR expression is also upregulated directly by aldosterone [106]. Usually, atrial fibrosis is also found in AF and it is mediated by the profibrotic effect of aldosterone. High aldosterone induces AF, hence blocking MR may prevent AF. Several clinical studies reported the preventable effects of MR blockers on AF [107,108]. One study evaluated the antiarrhythmic effect of spironolactone, an MR blocker, under β-blocker treatment of patients with recurrent AF episodes and identified the preventable effect of spironolactone on AF [107]. Eplerenone, another MR blocker, reduces the new onset of AF in patients with low ejection fraction and mild symptoms [108]. These findings suggest that MR antagonists suppress the incidence of AF. 

## 6. Aldosterone and Renal Function

Excess aldosterone leads to CKD through tissue inflammation, injury, glomerulosclerosis, and interstitial fibrosis. In an animal model, rats fed with a high-salt diet along with chronically administered aldosterone exhibited intrarenal vascular and glomerular sclerosis, as well as proteinuria [109]. In a clinical study, patients with PA showed higher serum creatinine levels and lower glomerular filtration rates than those with EH [110]. This study also demonstrated that lower glomerular filtration rates were predicted by initial potassium and plasma aldosterone concentrations and the presence of hypokalemia [110]. Moreover, 24 h urinary albumin excretion was significantly higher in patients with PA than in those with EH [111]. A recent prospective observational study demonstrated that individuals with CKD and high aldosterone levels in serum had an increased risk of CKD progression regardless of also having diabetes mellitus [97]. In another study, MR antagonists improved the urinary albumin–creatinine ratio in patients with diabetic nephropathy, who were receiving a treatment based on an angiotensin-converting enzyme inhibitor (ACEI) or an angiotensin receptor blocker (ARB) [112]. Interestingly, most patients on ACEI or ARB treatments are observed to present a phenomenon named “aldosterone breakthrough,” in which the plasma aldosterone concentrations return to, or even exceed, pretreatment levels, following an initial reduction [113]. An aldosterone breakthrough is associated with more severe proteinuria and a faster decline in renal function [114]. Thus, the patients on ACEI/ARB may need MR antagonists to block aldosterone breakthrough. 

The harmful effect of aldosterone on the kidney occurs through the non-epithelial MRs and can arise independent of the effect of aldosterone on blood pressure [115]. The MR detrimental activation for renal damage is linked to several molecular mechanisms, such as inflammation, oxidative stress, and vascular injury [115]. Mainly, inflammation and fibrosis play central roles in the pathophysiology of renal injury through MR activation via aldosterone. MR activation can decrease eNOS activity and uncoupling, resulting in impaired vasodilation [116]. Moreover, eNOS uncoupling increases hydrogen peroxide production and activates the NF–κB pathway, leading to inflammation and fibrosis [117]. Impaired NO activity enhances proteinuria and accelerates innate immune system activation, which causes tubulointerstitial injury [118]. Aldosterone-infused rats showed increased renal expression levels of proinflammatory cytokines from macrophages, which are key mediators of MR-induced injury [119]. Aldosterone also induced collagen synthesis in cultured fibroblasts and glomerular mesangial cells [120,121]. Moreover, MR antagonists prevent progressive kidney dysfunction by reducing blood pressure and decreasing profibrotic and inflammatory mediators [122]. Spironolactone treatment in patients with refractory hypertension reduced collagen synthesis independent of blood pressure reduction caused by other medications [123]. Eplerenone treatment reduced aldosterone-induced kidney fibrosis, as well as monocyte chemotactic protein 1(MCP-1), ICAM-1, and TGF-β expression, in mice [124]. One clinical study also reported that MR antagonists longitudinally slowed down the reduction of eGFR slope with enough MR blockage [125]. 

Patients treated with the MR antagonist and ACEI/ARB showed lower albuminuria compared with those treated with ACEI/ARB alone [126]. MR antagonists may contribute to the beneficial effect on the kidney by reducing proteinuria and suppressing CKD progression. As higher aldosterone levels are associated with CKD progression than lower aldosterone levels, aldosterone may enhance renal damage; therefore, suppression of these adverse effects can prevent kidney dysfunction. 

Aldosterone may have a direct, harmful effect on the kidney and induce hypertension. The physiological effects of aldosterone also increase urinary excretion of potassium. In PA, hypokalemia occurs due to excess aldosterone [127]. Historically, hypokalemia has been known to induce kidney dysfunction [128]. The renal disturbance due to excess aldosterone may be caused by the direct aldosterone effect and the subsequently induced hypokalemia. The renal dysfunction caused by chronic hypokalemia is hypokalemic nephropathy. The pathogenesis of hypokalemic nephropathy is assumed to increase vasoactive mediators, which leads to renal vasoconstriction, decreased medullary blood flow, and impaired renal angiogenesis [129]. Some of the histological findings that can lead to progressive renal dysfunction are the presence of intracytoplasmic vacuoles in renal tubular cells, chronic inflammation, and interstitial fibrosis [130].

## 7. Conclusions

This review aimed to collect and put into perspective recent available research findings regarding the effects of aldosterone on metabolic processes and the cardiovascular system. Excess aldosterone alters both metabolism and RAS pathophysiology, which leads to increased levels of inflammatory and fibrotic mediators, as well as the occurrence of inflammation, oxidation, and fibrosis in the cardiovascular and renal systems. These events cause adverse clinical outcomes, such as CAD, AF, CKD, and even the death of the patients. Moreover, aldosterone changes vascular stiffness and induces vascular thrombosis, and tends to promote the occurrence of cardiovascular ischemic events. Thus, it should be considered that aldosterone-mediated cardiovascular and renal damage may be more severe than expected. Therefore, excess aldosterone, especially PA, needs suitable management to prevent worse outcomes. 

## Figures and Tables

**Figure 1 ijms-24-05370-f001:**
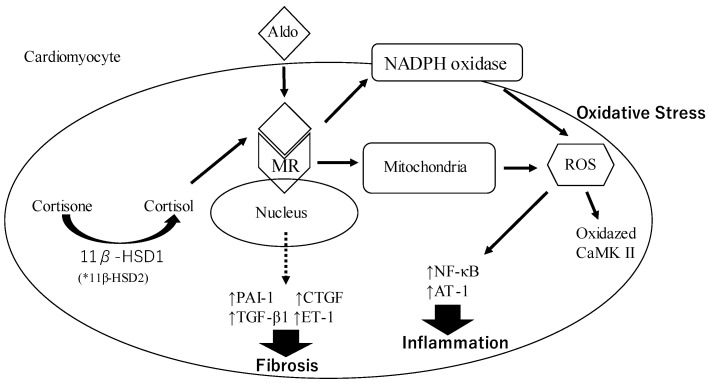
The mechanism underlying oxidative stress, inflammation, and fibrosis through the genomic pathway by aldosterone. * In the renal nephron and colon, 11β-HSD2 converts cortisol to the inactive form of 11-ketometaabolite; therefore, cortisol is not associated with genomic pathological effects of aldosterone in the kidney.

**Figure 2 ijms-24-05370-f002:**
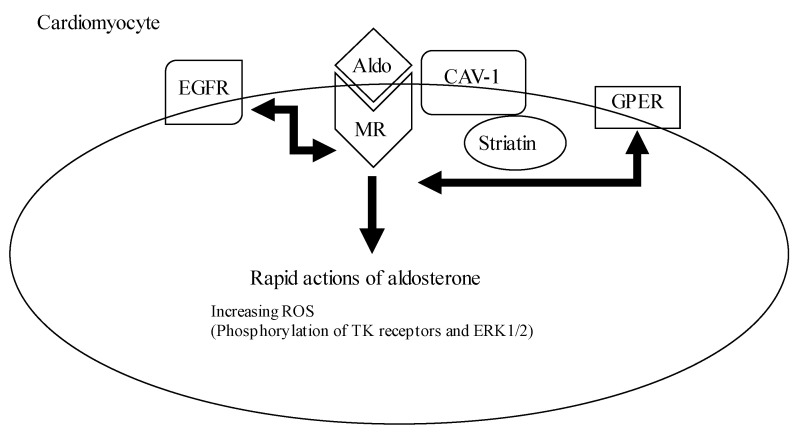
The mechanism underlying the pathological non-genomic effect of aldosterone in cardiomyocytes.

**Table 1 ijms-24-05370-t001:** Reports of cardiovascular and renal disease by aldosteronism between 2018 and 2022.

Author (Year)	Country	Study Design (Number)	Findings
Cardiovascular disease			
Youichi Ohno (2018) [84]	Japan	Retrospective cross-sectional study (n = 2582)	Patients with PA have a higher prevalence of CVD than those who are age- and sex-matched with EH.
Min-Tsun Liao (2019) [87]	Taiwan	Prospective observational and cross-sectional study (n = 336)	In PA patients, the relationship between eGFR and LVMI is not linear.
Yi-Yao Chang (2019) [88]	Taiwan	Prospective observational study (n = 249)	Adjusting by propensity score matching between the patients with PA and EH, patients with PA had worse LV diastolic function than patients with EH.
Chien-Ting Pan (2020) [89]	Taiwan	Retrospective matched case-control study (n = 11,010)	The patients with PA who underwent adrenalectomy had a lower incidence of NOAF compared with those with EH.
Jacopo Burrello (2020) [90]	Italy	Retrospective observational study (n = 5100)	PA and hypokalemia are associated with an increased risk of cardiovascular events.
Youichi Ohno (2020) [91]	Japan	Retrospective observational study (n = 1186)	Nadir PAC in the patients with PA after confirmatory tests is associated with LVMI, not the basal aldosterone level itself.
Teresa M Seccia (2020) [92]	Italy	Prospective observational study (n = 411)	Unexplained atrial fibrillation in the hypertensive patients shows a high prevalence of PA.
Jinbo Hu (2021) [93]	China	Cross-sectional study (n = 5521)	Patients with renin-independent aldosteronism are more closely associated with CVD risk than those with renin-dependent aldosteronism.
Tao Wu (2021) [94]	China	Prospective observational study (n = 70)	Compared with patients with EH, patients with PA have a higher degree of LV hypertrophy.
Arleen Aune (2021) [95]	Norway	Cross-sectional study (n = 198)	Patients with PA have a higher prevalence of LV hypertrophy both in women and men, compared to EH.
Lin Gun (2022) [96]	China	Retrospective cohort study (n = 3173)	Higher plasma aldosterone concentration is associated with increased risk of CVD and all-cause mortality in the hypertensive patients, even independent of OSA and PA.
Chronic kidney disease			
María Fernández-Argüeso (2021) [2]	Spain	Case-control study (n = 100)	Compared with patients with EH, patients with PA have a higher prevalence of CKD at the time of diagnosis.
Ashish Verma (2022) [97]	United States of America	Prospective observational study (n = 3680)	Regardless of concomitant diabetes, high serum aldosterone levels in the serum of patients with CKD are independently associated with an increased risk for CKD progression.

Abbreviations: PA: primary aldosteronism, CVD: cardiovascular disease, EH: essential hypertension, NOAF: new onset atrial fibrillation, OSA: obstructive sleep apnea, PAC: plasma aldosterone concentration, LVMI: left ventricular mass index, eGFR: estimated glomerular filtration rate, CKD: chronic kidney disease.

## Data Availability

Not applicable.

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
