# Peer review of "The Effect of Aldosterone on Cardiorenal and Metabolic Systems"

_ijms, 2023, doi:10.3390/ijms24065370_

Round 1

Reviewer 1 Report

In the manuscript (MS) entitled “The Effect of Aldosterone on Cardiorenal and Metabolic Systems” Hiromasa Otsuka et al. review the current evidence on aldosterone and its biological effects in the context of primary hyperaldosteronism and other cardiovascular and renal diseases. The MS follows typical structure for a review paper: it starts with presentation of physiology of aldosterone synthesis and release, mineralocorticoid receptor dependent and independent effects of aldosterone; then more detailed information on pathological actions of aldosterone at cellular/tissue level is provided; eventually, the role of aldosterone in specific pathologies is discussed.

The MS is generally well written and concise, with references to relevant and current literature. In my view the MS provides a valuable overview of aldosterone pathophysiology, especially that the authors go beyond typical clinical context (such as heart failure or hypertension) and discuss recent findings related to the role of aldosterone in thrombosis a atrial fibrillation.

Nonetheless, I have some questions/suggestions listed below:

Major points:

1. Selectivity for binding of steroid hormones (mineralo- and glucocorticoids) to mineralocorticoid receptors (MRs) depends on expression of 11beta-hydroxysteroid dehydrogenase (11beta-HSD) type 1 and type 2. Specifically, in vivo 11beta-HSD1 shows mostly reductase activity, thus it activates ketoanalogues into cortisol, and 11beta-HSD2 is unidirectional dehydrogenase that decomposes cortisol and prevents activation of MRs by glucocorticoids. Thus, in the section “2. Aldosterone Synthesis and Physiology” the role of type 1 and type 2 11beta-HSDs in maintaining selective stimulation of MR by aldosterone in some tissues (for example kidney) and non-selective stimulation of MR by cortisol in other tissues (for example immune cells or adipocytes) should be mentioned. There are several recent reviews that discuss this issue and should be referenced.

2. Please, note that in addition to GEPR1, the non-genomic effects of aldosterone are also associated with other receptors, such as EGFR, PDGFR and IGF1R and angiotensin AT1 receptor. I suggest including this information with relevant references (for example see: Ruhs S, Nolze A, Hübschmann R, Grossmann C. 30 YEARS OF THE MINERALOCORTICOID RECEPTOR: Nongenomic effects via the mineralocorticoid receptor. J Endocrinol. 2017 Jul;234(1):T107-T124).

3. The section on aldosterone and atrial fibrillation is very interesting. I suggest including additional references to clinical trials that showed beneficial effects of MR blockade with either spironolactone or eplerenone in patients with atrial fibrillation:

Dabrowski R, Borowiec A, Smolis-Bak E, Kowalik I, Sosnowski C, Kraska A, Kazimierska B, Wozniak J, Zareba W, Szwed H. Effect of combined spironolactone-β-blocker ± enalapril treatment on occurrence of symptomatic atrial fibrillation episodes in patients with a history of paroxysmal atrial fibrillation (SPIR-AF study). Am J Cardiol. 2010 Dec 1;106(11):1609-14. doi: 10.1016/j.amjcard.2010.07.037. Epub 2010 Oct 14. PMID: 21094362.

Swedberg K, Zannad F, McMurray JJ, Krum H, van Veldhuisen DJ, Shi H, Vincent J, Pitt B; EMPHASIS-HF Study Investigators. Eplerenone and atrial fibrillation in mild systolic heart failure: results from the EMPHASIS-HF (Eplerenone in Mild Patients Hospitalization And SurvIval Study in Heart Failure) study. J Am Coll Cardiol. 2012 May 1;59(18):1598-603. doi: 10.1016/j.jacc.2011.11.063. PMID: 22538330.

Minor:

1. Abbreviation "MR" is not explained in the body of the text, only in the figure caption.

2. There are several small issues related to formatting of References. Here are the ones I spotted, but the MS needs careful proofreading.

Line 424 – digit 9 is lost in the numbering of references

Line 548 – reference 70 is not aligned

Line 565 – reference lacks the full stop after 78

Reviewer 2 Report

Otsuka et al. reported the review article entitled ‘the effects of Aldosterone on Cardiorenal and metabolic systems”. Although the description is interesting, most of them are superficial. It is unclear whether the authors want to describe the pathophysiological role of aldosterone or the specific condition primary aldosteronism.

11b-hydroxysteroid dehydrogenase type 2 (11b-HSD) has a significant role for the pathophysiological effects of aldosterone on MR. Does 11b-HSD have no role for aldosterone action to the heart? Fig.1 has to be revised. The review by Buonafine H. et al (AJH 2018)would be helpful.

The authors describes that the MR detrimental activation for renal damage is related to several molecular mechanisms such as inflammation, oxidative stress and vascular injury by citing reference 103. This is difficult to understand for clinician and is not helpful for the treatment of patients with hypertension.

Aldosterone via MR influences not only ENaC in principal cells but also H-ATPase and H-K-ATPase in the intercalated cells to regulate sodium and potassium homeostasis, respectively. Such a primitive description of aldosterone action in the kidney is required.

Reviewer 3 Report

This manuscript reviews the deleterious effect of elevated aldosterone in the kidney and cardiovascular system. Several mechanisms of the actions of aldosterone are highlighted including oxidative stress and inflammation, as well as relevant genomic and non-genomic effects. The manuscript is well-written and highlights the importance of early interventions for patients with primary aldosteronism.

My only suggestion if to add more figures summarizing the different aspects of aldosterone function in the kidney and cardiovascular system, genomic and non-genomic effects of the hormone and the target tissues. 

Round 2

Reviewer 2 Report

The revised version of the manuscript improved very much by adding 11beta-HSD.  i have no further comments.